# Low-Cost Technologies Used in Corrosion Monitoring

**DOI:** 10.3390/s23031309

**Published:** 2023-01-23

**Authors:** Mahyad Komary, Seyedmilad Komarizadehasl, Nikola Tošić, I. Segura, Jose Antonio Lozano-Galant, Jose Turmo

**Affiliations:** 1Department of Civil and Environment Engineering, Universitat Politècnica de Catalunya, BarcelonaTech. C/ Jordi Girona 1-3, 08034 Barcelona, Spain; 2Department of Civil Engineering, Universidad de Castilla-La Mancha, Av. Camilo Jose Cela s/n, 13071 Ciudad Real, Spain

**Keywords:** corrosion monitoring, electrochemical sensors, data acquisition, data analysis, physical methods, structural health monitoring, low-cost sensors, atmospheric corrosion

## Abstract

Globally, corrosion is the costliest cause of the deterioration of metallic and concrete structures, leading to significant financial losses and unexpected loss of life. Therefore, corrosion monitoring is vital to the assessment of structures’ residual performance and for the identification of pathologies in early stages for the predictive maintenance of facilities. However, the high price tag on available corrosion monitoring systems leads to their exclusive use for structural health monitoring applications, especially for atmospheric corrosion detection in civil structures. In this paper a systematic literature review is provided on the state-of-the-art electrochemical methods and physical methods used so far for corrosion monitoring compatible with low-cost sensors and data acquisition devices for metallic and concrete structures. In addition, special attention is paid to the use of these devices for corrosion monitoring and detection for in situ applications in different industries. This analysis demonstrates the possible applications of low-cost sensors in the corrosion monitoring sector. In addition, this study provides scholars with preferred techniques and the most common microcontrollers, such as Arduino, to overcome the corrosion monitoring difficulties in the construction industry.

## 1. Introduction

Continuous maintenance and repair work are two vital factors to guarantee the necessary conditions of safety, economy, and functionality during the life cycle of infrastructures. However, the lack or inefficiency of these actions can significantly accelerate the deterioration of structures leading to more costly reparations. One harmful phenomenon that might jeopardize the structural integrity of structures is corrosion, which is a thermodynamically spontaneous process. In fact, the cost incurred from corrosion contributes more than the total damage caused by all natural disasters (such as earthquakes, floods, and storms) combined. The World Corrosion Organization (WCO) estimates that the annual cost of problems caused by corrosion worldwide reaches $US2.4 trillion, which is 3% of the world’s gross domestic product (GDP) [1]. In many developed nations, corrosion can cause structural damages with costs approaching 3.5% to 4.5% of the gross national product (GNP) [2]. For example, on 22 November 2013, corrosion was the main reason for the oil pipeline explosion in the city of Qingdao in China’s eastern Shandong province, as it thinned the pipeline’s wall, leading to the break and causing a substantial economic loss of $US123.9 million [3]. In addition to economic losses, corrosion might lead to structural collapses with consequent casualties. For instance, the Guadalajara sewer explosion in Mexico in April 1992 claimed the lives of 215 people and injured another 1500 [4]. In addition, a $US75 million financial loss was estimated. The incident was linked to a gas line’s degradation, which led to a gas leak into the nearby sewage main.

Moreover, corrosion is also responsible for the leakage of devastating pollution into the environment. For instance, on 12 December 1999; the tanker Erika sank off of France’s Brittany coast [4]. This type of catastrophe also caused a heavy oil spill globally in 1998, about 19,000 tons [5]. Another recent accident occurred during a severe rainfall event on 14 August 2018, when a 210 m portion of Ponte Morandi collapsed. Corrosion of the concrete reinforcements was listed as one of the primary causes of the collapse [6].

These disasters could have been avoided if regular corrosion detections had been conducted. Corrosion detection, in general, is the process of evaluating the relevant types of corrosion in a particular environment and identifying the critical variables impacting the rate of propagation, which is divided into corrosion inspection and corrosion monitoring [7]. Inspection typically refers to short-term “once-off” measurements performed in accordance with maintenance and inspection schedules, whereas corrosion monitoring refers to the measurement of corrosion damage or variables that accelerate the corrosion rate over a longer time period. Corrosion monitoring frequently includes an attempt to gain a deeper understanding of how and why the corrosion rate fluctuates over time. Therefore, nowadays, corrosion monitoring has been attracting considerable attention among different industries, such as oil and gas, civil infrastructures, and the marine industry as a means to guarantee their structures’ safe operation. Figure 1 illustrates the distribution of papers focused on corrosion monitoring from the SCOPUS database from the years 2004 to 2021. Analysis of this graph shows a constant interest in this field among researchers.

Corrosion monitoring is carried out using a set of detectors, data storage and transmission systems. The implemented system analyzes the detected data and assesses the corrosion state of the structure and its components [4]. Among different types of detectors, metal coupons are one of the most widely used and reliable methods for corrosion detection in different industries. Metal coupons are exposed to corrosive conditions in order to calculate the mean corrosion rate from the mass loss, mass gain, or coulometric decrease of corrosion products. However, this technique does not provide real-time measurements, and corrosion monitoring usually relies on electronic corrosion sensors or probes [8]. These electronic corrosion sensors or probes continuously or semi-continuously transmit corrosion information to the system. Afterwards, by using the proper corrosion management algorithm in the corresponding data acquisition system, not only the performance data but also a basis for life prediction of the under-study system are provided. Therefore, continued development of these electronic devices has a high level of importance, as corrosion monitoring is one of the most essential components in corrosion prevention and corrosion control. Figure 2 illustrates the scheme of a corrosion monitoring system, and, as can be seen, it contains three main parts: (1) a sensor or probe that is connected to a corrosive area in order to measure the events and changes of the environmental parameters; (2) a data acquisition system which is used to collect and transfer the data; and (3) a data processing and storing component where the collected data are analyzed and stored.

Corrosion is a highly complex process involving at least two phases. For instance, solid and liquid; solid and gas; solid, liquid, and gas; or even a solid and a first and second liquid might be involved in the corrosion process. Therefore, corrosion monitoring is a multidisciplinary task, and very often, two or more methods are needed in order to adequately manage the monitoring needs in a given situation. However, because almost all corrosion processes in nature take place in the aqueous phase, electrochemical detection and monitoring procedures are considered to be more efficient compared to physical methods [9]. In addition, electrochemical techniques have several advantages, including sensitivity to low corrosion rates, a short experimental period, and a well-established theoretical understanding [10]. Nevertheless, due to their high sensitivity and accuracy in corrosion measurements, monitoring for field applications is more complex than laboratory experiments. Electrochemical techniques that are appropriate for field applications should have two key properties. On the one hand, the methods must be dependable and straightforward, which is crucial for long-term corrosion monitoring. Numerous complex electrochemical techniques exist that are effective for understanding the kinetics and mechanism of corrosion. Nevertheless, they may not be applicable to field evaluation, because they are either too slow or too complex for the typical operator to utilize in the field. On the other hand, in order to minimize the negative impact on the materials being studied, the techniques used should be non-destructive.

Depending on different natural environments, such as corrosion in soils, atmospheric corrosion, fresh-water corrosion and seawater corrosion, corrosion monitoring instances are usually considered separately, and different electrochemical or physical techniques are used. For instance, in the case of corrosion in different industries (i.e., chemical processing, oil and gas production, power generation, civil infrastructures), different electronic systems are implemented [11]. Lately, with the development of technology the demand for adopting sensors in civil infrastructures and buildings for static or dynamic structural system identification (SSI) has increased. However, in order to implement sensors to monitor the corrosion in these structures, the atmospheric conditions are normally monitored. Although there are many methods available for monitoring metal corrosion, only some of them can be applied in the atmosphere [12]. Among all of the methods available, atmospheric corrosion monitoring (ACM), electrochemical impedance spectroscopy (EIS), electrochemical noise (EN), and electrical resistance (ER) probes stand out for their unique ability to monitor the real-time data of environmental corrosiveness [13]. ACM is also known as a galvanic corrosion sensor, which is composed of two different metal electrodes insulated from each other. One of them is fashioned from a noble metal, and the other one is formed from the metal of interest in order to create galvanic corrosion, and by measuring the galvanic current using a galvanometer or a zero-resistance ammeter (ZRA), the corrosion current and corrosion rate are measured [14]. EIS is a non-destructive monitoring technique that provides information about the corrosion progress and condition of protective coatings. EIS measures the amplitude and phase of the surface impedance at different frequencies. Traditionally, measurements are performed by employing a conventional electrochemical cell filled with an electrolytic solution so that, if a corrosion product layer coats the metallic surface, the current gets through owing to both an electronic and an ionic contribution. Since corrosion products, such as metallic oxides and hydroxides, have low electronic conductivity, their ionic contribution can become significant. Therefore, measuring the impedance at different frequencies enables the study of electrochemical reactions on the metallic surface and permits highlighting of the presence of cracks and porosity on the surface layer, both of which can act as starting points for localized corrosion attacks [15,16,17]. Although the use of EIS in atmospheric circumstances is constrained by the requirement for an electrolyte, it has recently been used in a number of atmospheric corrosion experiments [18,19]. Generally speaking, EN refers to the current or potential fluctuation that occurs on a corroding electrode. In this method, the current between two identical electrodes (ECN) and the potential difference between the two identical electrodes compared with the potential of a reference electrode (EPN) are measured simultaneously. The potential measurement (EPN) is related to the corrosion mechanism, whereas the current data (ECN) indicates the corrosion rate [20]. EN is the only electrochemical approach that is ideal for in situ measurements [21], straightforward to set up, and does not affect the system while conducting experiments [22]. ER probes have been widely used since the 1950s to track corrosion in soils, solutions, concrete, and atmosphere [23]. In short, a metallic sensor crafted from the same element as the materials of the structure under study is exposed to the corrosive environment and its resistance is monitored by detecting a potential drop while applying a small current. Primarily thin films are used for atmospheric systems due to their high sensitivity. The principle relies on the fact that the resistance of an element is a function of its resistivity and geometrical parameters (length, width, and thickness). Therefore, by tracking the resistance values, the values of the thickness loss due to corrosion and the corrosion depth (CD) will be acquired, and consequently, the corrosion rate, which is the slope of CD with respect to time, is monitored [5].

Table 1 provides an overview of the techniques mentioned for monitoring corrosion in the atmosphere. For a quick orientation, the sensitivity is indicated in general categories. Moreover, the main benefits and drawbacks of each method are presented.

The cost of commercial corrosion systems is a crucial determinant to their suitability for use in given services. Therefore, Table 2 presents a list of common corrosion monitoring devices with real-time updates available on the market for each technique, including their names, measurement ranges, and their retail prices in 2022.

As can be seen from Table 2, commercial corrosion monitoring devices are characterized by their high precision and their high price. In addition to these prices, the cost of data acquisition equipment (such as cables, a corrosion data logger, an inductance transmitter, and a corrosion management system) must be added. These needed auxiliary elements can increase the prices of the aforementioned sensors by up to an additional $US7000. Moreover, it is well established that corrosion detection and monitoring in outdoor applications is more complicated in comparison to laboratory experiments. Some areas that require ongoing development are probe/sensor design, data processing, and electrochemical instrumentation. In some circumstances, environmental noise is mixed with the observed data, thereby necessitating the use of appropriate signal processing technology to filter or even analyze this noise. In fact, because many corrosion systems are not steady-state, and as the corrosion potential may change, it is possible to apply transient techniques like electrochemical noise analysis (ENA) or other physical methods (image analysis [28,29,30], acoustic emission [31,32,33,34]). However, these systems are generally quite large and expensive, because they need to be equipped with complex auxiliary assets.

The high price tag on corrosion monitoring systems leads to their exclusive use with limited measuring points. These systems are typically used only in areas with the highest probability of corrosion, for example, inside petrochemical pipelines. However, in conventional structures, which are those in which many pathologies occur, the provision of corrosion monitoring systems is not usual, and insufficient attention has been paid to evaluating the progress of corrosion, especially in the case of atmospheric corrosion attacks in civil infrastructure. Additionally, this high price tag is a barrier to the long-term monitoring of structures on a low structural monitoring budget.

Therefore, the demand to use low-cost probes and open-source platforms for data acquisition systems by implementing various techniques to improve their accuracy compared to high-cost precision systems has attracted considerable attention for structural assessments [35,36,37]. This includes improvement such as increasing the density of measurement points as well as making it possible to implement long-term monitoring systems in order to reduce the uncertainty of the obtained data [38]. Low-cost sensor (LCS) usage for many aspects of structural health monitoring is evident in the literature due to the rapid advancement of micro sensor technology and microcontrollers (such as Arduino and Raspberry Pi) [39,40]. The monitoring of acceleration [41,42,43], tilt [44], temperature [45,46], humidity [47], and corrosion [19] are a few examples of this research. Komarizadehsasl et al. [48] reported that, with a proper code and the right mode to exploit the features and potential of inexpensive electronics, it is possible to obtain valuable data that can provide useful insights for structural health monitoring applications. In this work, a high-precision low-cost inclinometer was developed through the fusion of multiple microelectromechanical systems (MEMS). There are also various review papers in the literature concerning the use of low-cost sensors (LCSs) in structural health monitoring applications. Mobaraki et al. [49] reviewed the application of some low-cost sensors for the measurement of temperature, humidity, and airflow for building monitoring.

For corrosion monitoring systems, there are several literature reviews concerning different electrochemical methods used both for in situ and laboratory assessments [50], corrosion monitoring in atmospheric conditions [13], different techniques for monitoring the corrosion of steel rebar in reinforced concrete structures [51], using fiber optic sensors for the assessment of rebar corrosion in RC structures [52], and the advancement of a sensor capable of corrosion detection [53]. However, a systematic literature review on the use of LCSs and inexpensive technologies as low-cost solutions for corrosion monitoring applications is lacking. Therefore, in order to fill this gap, the current review offers a thorough, up-to-date analysis of the state of the art for using low-cost digital technologies for sustainable corrosion monitoring by examining the SCOPUS database. Additionally, the authors pay close attention to the presentation of many elements of the main fields of corrosion monitoring (such as techniques and ambient conditions) and the most popular sensors used in each technique.

The current review is divided into four sections. The primary topic of discussion in this study is the research on in situ measurements of atmospheric corrosion. The background and research significance of corrosion detection and monitoring are briefly introduced in Section 1. The rest of the paper is organized as follows. The data sampling technique for low-cost corrosion technologies is described in Section 2. In addition, the approach for processing data from inexpensive monitoring equipment is presented. The results are presented in Section 3. Specifically, Section 3.1 provides a thorough discussion of the many domains of corrosion monitoring as well as the reviewed articles. The distribution of the publications among different countries and industries are also reviewed in this section. Section 3.2 reviews the use of low-cost corrosion monitoring systems using electrochemical methods. In addition, the open-source microcontrollers discussed in the literature are evaluated, and the ones recommended for use in the installation of inexpensive monitoring devices are described. Section 3.3 describes and compares the use of low-cost technologies to detect and monitor the physical parameters induced by corrosion. The conclusions of the current literature review are presented in the final section.

## 2. Materials and Methods

This section describes the sources of the data, the evaluation techniques, and the classifications of low-cost technologies used for corrosion monitoring. Two main fields of corrosion monitoring in the industry are associated with (1) electrochemical methods and (2) physical methods. These two fields could provide and cover a structure’s corrosion monitoring for atmospheric corrosion. Measurements made in the field are the next primary discipline of this review. This paper addresses the publications in the SCOPUS database between the years 1986 and 2021. The current review was carried out in November 2022. The initial search technique used for this systematic review paper consists of three major components. The first stage is meant to provide an initial set of articles found by employing a certain search algorithm using the Boolean operator AND (Figure 3). The obtained articles in the first stage were then filtered further in the next step in order to eliminate duplicates and irrelevant ones. Following this process 209 articles were deemed inappropriate because they lacked information on corrosion monitoring or did not clearly cover low-cost features. It shall be highlighted that in this study, the authors only considered peer-reviewed papers written in English. This exclusion was accomplished using the “AND NOT” command, and the final outcome was 38 using electrochemical methods and 52 articles implementing physical methods to track and monitor the essential parameters of corrosion across all industries.

In the final step, a detailed review of the selected articles was conducted in terms of defining the main methods featured in the selected articles, assessing their distribution in different countries around the world, analyzing their contribution in different industries, and finally, evaluating the types of sensors and technologies used for the installation of the low-cost corrosion monitoring system.

There are several methods available in the literature to detect corrosion and monitor its essential parameters for different type of structures. However, the principal aim of this article is to present comprehensive information about low-cost technologies implementing electrochemical methods in this field, and to do so, only corrosion detection methods that are compatible with the intended inexpensive electronics are reviewed in this paper. Therefore, this review can be used as a thorough reference for scholars interested in low-cost corrosion monitoring. Moreover, this review provides comprehensive information about low-cost technologies measuring physical parameters to track corrosion progress as well as the preferred data acquisition systems and their important parameters for use in low-cost corrosion monitoring applications. Table 3 provides a summary of the corrosion monitoring methods introduced so far in the literature and their designated categories.

Recent advances in data processing technology, along with an ever-increasing demand for online structural health monitoring, have resulted in the development of more efficient monitoring systems. One of the most essential features of current low-cost monitoring systems is that they run open-source software. This enables the users to access the library source of the low-cost sensors, thereby giving them the chance to improve and develop their own algorithms in the post-processing stage [54]. However, the vast majority of the commercial corrosion monitoring devices on the market have already been programmed by the associated companies, and they are based on closed source platforms. This implies that users have no access to inspect and develop the software, which represents a huge drawback for researchers interested in this field. Therefore, this paper provides insight to the scholars for choosing the most desirable low-cost open-source monitoring system for their project by reviewing the most-preferred options in the literature.

## 3. Results

### 3.1. Overview of the Results

The first study found in the SCOPUS database dealing with the idea of low-cost corrosion monitoring is dated to 1984 [55]. However, this paper is not included in the statistical analysis of this research, as it only introduced the use of various types of probes with portable devices to reduce the risk and the cost of the corrosion monitoring procedure in the oil and gas industry. In fact, the actual first paper that focused on the use of a low-cost corrosion monitoring device was dated to 1986. Knuutila et al. were the first researchers to describe the use of a low-cost microcomputer compatible with electrochemical systems for polarization and cyclic voltammogram measurements [56]. Since then, the number of yearly publications on the subject was limited until 2008, when a significant increase was observed (Figure 4). A second relative rebound can be seen between 2008 and 2012. It was observed that more than 54% of all the publications over a 36-year period (1986–2021) were condensed between 2012 and 2021, highlighting the importance of technology´s growth on the development of new low-cost electronics. It can be seen from Figure 4 that the majority of the authors investigated low-cost approaches using physical methods (almost 58%), whereas only 42% of the articles introduced economic developments by employing electrochemical methods. In addition, it was observed that from 1986 until 2007, the number of papers associated with the use of electrochemical methods for corrosion monitoring was quite few (23.7%), and more than 76.3% of the articles were published in the last 14 years (from 2008 to 2021). This rebound in projects concentrating on the use of physical methods for corrosion monitoring applications occurred within the same amount of time as the number of papers that contributed more than 78.9% in the last 14 years, and only 21.1% of them were published in the years 1986 to 2007.

Figure 5 illustrates the number of papers by country. It an be observed that, in terms of the publication numbers, the countries that put forth the largest contribution to the use of low-cost technologies for corrosion monitoring are the United states (43 papers), the United Kingdom (7 papers), Italy and China (6 papers each), Canada (5 papers), and Australia and Portugal (3 papers each), followed by Spain, Mexico, India, and Germany (2 papers each). In total, researchers from 26 countries around the world contributed to investigating and introducing the use of new low-cost systems in different industries in order to overcome corrosion monitoring’s high price tag problem in all 90 of the reviewed publications.

Figure 6 shows the contribution of the reviewed publications to different industries. It can be seen that 51% of the researchers devoted their studies to developing low-cost solutions for monitoring the corrosion parameters of civil structures. The second most-interesting industry for corrosion monitoring using low-cost technologies is the oil and gas industry, for which the most pipeline pathologies have occurred, and 18% of the publications implemented low-cost solutions to overcome the high price tag of conventional corrosion monitoring systems in this industry. The analysis in Figure 6 shows that the monitoring of artifacts is considered to be a vital area among researchers for developing low-cost monitoring techniques for tracking and controlling the corrosion process of these valuable assets. The rest of the industries for which researchers developed low-cost corrosion monitoring systems are shown in Figure 6.

### 3.2. Electrochemical Methods Used for Corrosion Monitoring

In this section, the reviewed articles associated with the use of electrochemical methods for low-cost corrosion monitoring techniques are presented. This first group of papers can be classified into four separate categories. The first category is a group of publications that devoted their research to the application of EIS in the corrosion field and especially to the development of low-cost systems compatible with this method. In the second category, authors utilized linear polarization resistance (LPR)-based devices to remotely track corrosion potential in their studies. In Section 3 and Section 4, Galvanic and EN methods were respectively used to measure and monitor electrochemical parameters for low-cost corrosion monitoring. Figure 7 shows the distribution of the reviewed publications in SCOPUS associated with the above-mentioned methods.

It can be seen that among the four groups, the highest number of papers dealt with the application of the EIS method for the measuring and monitoring of electrochemical parameters involved in the corrosion process. This category contained 58% of the publications from 1986 to 2021. Analysis of Figure 7 proves that EIS is considered among authors to be the most reliable method of monitoring the coating condition and the corrosion progress for their projects [57,58,59]. It also shows that the distribution of the publications dealing with LPR as their main method was almost constant during the years (from 2007 to 2018) with no significant changes, representing 34% of the total publications in the field of low-cost corrosion monitoring. Although EN and galvanic are considered to be useful electrochemical methods in the corrosion industry, they have not attracted the attention of researchers for low-cost monitoring applications, and they only contributed 3% and 5% (respectively) of the entire number of publications. The main reason for this low participation is their complicated data interpretation and the short life span of their sensors, respectively [13].

#### 3.2.1. Electrochemical Impedance Spectroscopy

This subsection provides detailed information on the reviewed publications that introduced low-cost corrosion monitoring systems based on the EIS principle. EIS is one of the most commonly used methods in the industry that enables researchers to study and monitor the coating conditions of metallic structures. However, the corrosion rate measurements produced by EIS are not so simple, and it is normally considered to be a time-consuming method due to the requirement of recording data at various frequencies [60]. In addition, EIS measurements require 3 electrodes as well as an electrolyte connection, which makes mostly limits this method to laboratory use. Therefore, attempts to develop low-cost portable EIS-based devices are considered to be highly valuable among various industries. These attempts are focused either on the development of the sensing component (low-cost electrodes or sensors) or the data acquisition implement (use of a low-cost microcontroller).

Licina et al. [61], Thien et al. [62], Park et al. [63], Yu et al. [64], Thien et al. [65], Abdur Rahman et al. [66], Mejía-Aguilar et al. [67], Azhari et al. [68], Friedersdorf et al. [69], Andrews et al. [70], Nazir et al. [57], Bansal et al. [71], Li et al. [72], Corva et al. [73], Chowdhury et al. [74], and Strachotová et al. [75] all employed new technologies (such as piezoelectric materials and advanced nanocomposites) for the sensing section of their EIS-based corrosion monitoring systems in order to obtain low-cost and low-power features. For instance, Yu et al. [64] used a piezoelectric wafer active sensor (PWAS) for the detection of coating thickness and the monitoring of the corrosion condition of metallic plates and pipes with a price of $US10 for each sensor.

Thien et al. [62] developed a low-cost active-sensing-based diagnostic system using flexible micro-fiber composite (MFC) patches as both the sensors and the actuators in order to detect cracks or corrosion damage along the pipeline structures by recording the impedance responses. Figure 8 illustrates that only 27% of the authors—such as Carullo et al. [76], Davis et al. [77], Angelini et al. [19], Matsiev [78], Grassini et al. [79], and Sebar et al. [80]—focused on the use of open-source microcontrollers (such as Arduino and AD5933) in the development of their reliable low-cost EIS-based data acquisition systems. Among the reviewed microcontrollers that are able to perform impedance measurement through various frequencies, Arduino board—which costs less than $US50—stands out for its unique benefits, such as ease of use, active user community and various available libraries due its open-source feature. Angelini et al. [19] and Grassini et al. [79] proposed a simple low-cost solution based on a commercial Arduino board coupled with a logarithmic amplifier to perform impedance measurements in the frequency range of 0.01 Hz to 100 kHz. Sebar et al. [80] proposed a novel low-cost device that measured EIS using a simple Teensyduino board in the frequency range of 0.01 to 10,000 Hz without need of an additional analog amplifier.

#### 3.2.2. Linear Polarization Resistance

LPR is one of the most commonly used methods for corrosion monitoring in field applications. In this method, a very small polarization potential (usually ± 30 mV) above and below the corrosion potential is applied to the sample. The current response recorded over this short range near the corrosion potential is linear. As a result, the polarization resistance (R_P_) is defined as the slope of this current–potential curve, which is constant. Under certain conditions R_P_ is inversely related to the instantaneous corrosion rate, according to the Stern–Geray equation [81]. In fact, polarization techniques such as EIS and LPR are considered highly accurate with easy data interpretation. Hence, many investigators have tried to develop low-cost monitoring solutions through the use of either open-source microcontrollers or developed sensing elements. In Figure 9 it is shown that 54% of the publications that implemented LPR method focused the development of their data acquisition system on using open-source microcontrollers, while 46% of the articles addressed the use of low-cost sensors.

For instance, Niblock et al. [82], Wood et al. [83], Abu Yosef et al. [84], Rafezi et al. [85], Perveen et al. [86], and Ahmad et al. [87] employed low-cost passive wireless sensors such as piezoelectric sensors for the detection of corrosion in pipelines and in concrete by taking advantage of the LPR technique. At the same time, Knuutila et al. [56], Arpaia et al. [88], Mugambi et al. [89], Kwan et al. [90,91], Samoilǎ et al. [92], and Degrigny et al. [93] devoted the main objective of their research to developing open-source hardware systems with low-cost and easy-to-use instrumentation for in-field measurements. For example, Samoilǎ et al. [92] presented virtual instrument technology made up of a low-cost but powerful Cypress PSoC microcontroller based on the polarization resistance technique.

#### 3.2.3. Galvanic or Atmospheric Corrosion Monitoring

In the late 1950s, the first attempts to quantify atmospheric corrosion using galvanic sensors were made. An ACM system is composed of two different metal electrodes insulated from each other. One of them formed from is a noble metal, while the other is crafted from the metal of interest in order to create galvanic corrosion, and by measuring the galvanic current using a galvanometer or a zero-resistance ammeter (ZRA), the corrosion current and corrosion rate are measured. Figure 10 shows a schematic illustration of an ACM sensor used to monitor automotive corrosion. However, due to the fact that galvanic coupling accelerates the corrosion process, which results in a shorter life span of the sensors and unclear data interpretation in harsh environments, researchers have not found this technique interesting for the development of low-cost monitoring solutions. Nevertheless, Castro-Borges et al. [94] proposed a reasonable, low-cost solution for the detection and monitoring of the corrosion performance of concrete columns after localized repairs using galvanic sensors to measure the electrochemical parameters. Mierau [95] proposed a remote monitoring solution for corrosion monitoring at airports, solving the access and field-diagnostic issues. The proposed low-cost PC-based control system, which employs galvanic sensors, provides corrosion engineers with a remote monitoring system with real-time updates.

#### 3.2.4. Electrochemical Noise

EN is a useful electrochemical method for investigating and monitoring corrosion processes such as pitting and stress corrosion cracks’ growth, microbial corrosion, and uniform corrosion. The measuring mode, the surface area of the working electrodes, the electrolyte resistance, and the symmetry of the electrode system all have a significant impact on the EN results. In addition, depending on the measurement mode, this technique can be used for in situ measurements employing either three-electrode or two-electrode setups [96]. Although the EN method has numerous benefits, such as localized corrosion detection because of its complex and unclear data interpretation, very little attention has been paid to using it to implement new technologies for low-cost monitoring developments. However, Arellano-Pérez et al. [27] developed a portable device for measuring and monitoring corrosion parameters such as electrochemical potential (EP) and electrochemical current (EC) based on EN signals. In this research, the corrosion type and rate, a probe made of three identical electrodes (6061-T6 aluminum) was constructed in order to evaluate. Furthermore, the proposed low-cost portable device includes a signal amplifier, a physical filter, and an analog-to-digital convertor (ADC), which enables it to measure weak signals and analyze the corrosion variables in different materials. Figure 11 illustrates both the three-electrode probe and the proposed low-cost portable device.

### 3.3. Physical Methods Used for Corrosion Monitoring

In this section low-cost approaches for corrosion monitoring implementing physical methods are reviewed. Controlling and tracking the physical parameters of the corrosion progress—such as metal thickness loss, crack propagation, and physical pathologies—are the main features of this method [97]. Unlike the electrochemical methods, these techniques do not directly measure the corrosion parameters such as corrosion potential or corrosion current. However, depending on the requirements of the projects, these techniques provide the possibility of measuring and tracking the physical changes caused by corrosion, and furthermore, calculating the corrosion rate. Therefore, in this section, the existing low-cost technologies for the measurement of the physical changes to structures caused by corrosion are reviewed. The reviewed publications are categorized into six distinct groups. The first category is a group of researchers that used ER as their principal method of measuring and monitoring corrosion rate using low-cost electronics. The second group of authors profited from radio frequency sensors (RF) and their unique features in their respective projects aimed at developing inexpensive remote corrosion monitoring systems. In the third group are authors who employed optical fiber (OF) sensors as their principal physical detection method for low-cost corrosion monitoring applications. The fourth group of authors focused on adapting ultrasonic waves (UW) to introduce their novel low-cost remote technologies for monitoring the coating and internal corrosion condition of structures in different industries. Authors from Group 5 implemented the electromagnetic induction (EMI) method to track physical changes using passive wireless sensors. Finally, in Group 6, authors addressed a novel technique taking advantage of image recognition (IR) technology to develop low-cost image sensors for tracking the progression of corrosion in their projects.

Figure 12 illustrates the rise in the number of publications associated with low-cost corrosion monitoring using physical methods that were published over the period from 1998 to 2021 and found in the SCOPUS database. It can be seen that researchers chose a more balanced distribution of physical methods suitable for low-cost corrosion monitoring applications as compared to the electrochemical approaches mentioned before. Nevertheless, among the six groups, the majority of the authors employed ER sensors and RF-based methods to detect physical changes (such as strain and thickness loss) and monitor them using inexpensive wireless technologies. These two groups contain 25% and 23% of the publications from 1998 to 2021, respectively. Analysis of Figure 12 shows that 75% of the articles focusing on the use of RF-based sensors were published in the past 10 years. At the same time, the least attention was paid to the use of IR for detecting and tracking structural pathologies. This technique has only been investigated in publications in recent years, with 100% of the articles having been published in the last 6 years. In fact, the developments in camera sensors and artificial intelligence (AI) technology in recent years have made them more accessible and thus more affordable for the development of low-cost corrosion monitoring systems [98]. Figure 12 also illustrates that the distribution of publications employing OF sensors and UW for low-cost corrosion monitoring solutions has been almost constant through across the years 1998 to 2021. However, low-cost monitoring solutions based on the EMI technique have been only reviewed by authors over a 10-year period between 2006 and 2016.

#### 3.3.1. Electrical Resistance

Low-cost sensors based on the ER principle have been widely utilized to monitor corrosion in various industries since 1998. As shown in Figure 13, these sensors generally contain two parts: a reference element, which is protected by a coating layer, and an exposed element. The resistance of the sensing element is measured by comparing its potential drop to the reference element, which is proportional to the loss in metal thickness experienced by the sensing element for a uniform corrosion mechanism. Furthermore, these values are used to calculate the accurate corrosion rate over time. Due to the simplicity and high precision of this technique, many scholars such as Taylor [99], Hautefeuille et al. [100], McCarter et al. [101], Holst et al. [102], Materer et al. [103], Halabe et al. [104], Hurley et al. [105], and Corva et al. [106] have introduced the use of low-cost ER sensors in their projects. Stromment et al. [107] introduced FSM (the field signature method) for the first time in 1998, which is an economical solution based on the ER principle for internal corrosion and crack monitoring in pipelines and vessels. In that study the authors combined non-destructiveness, high sensitivity, and real-time updateability with the ability to cover large areas and no maintenance requirements as the main features of their technique in order to develop their low-cost corrosion monitoring system. Andringa et al. [108,109] and Pasupathy et al. [110,111] demonstrated in their publications the use of sacrificial transducers as sensing elements for monitoring corrosion in concrete. This passive wireless sensor, which is ideal for in situ measurements in inaccessible locations, contains a resonant inductor–capacitor circuit with a resistive transducer that is exposed to the corrosive ambient in order to perform resistance measurement based on the ER technique.

#### 3.3.2. Radio Frequency

Radio-frequency identification (RFID) sensors are generally employed for inventory management and product tracking, and their application for corrosion monitoring is a relatively new concept established in the previous decade. However, in recent years numerous authors have addressed applications of this method in the field of corrosion monitoring. Bouzaffour et al. [112] introduced an innovative embedded UHF RFID (ultra-high-frequency RFID) sensor in concrete in order to detect the mass loss of the steel and monitor its corrosion process. El Masri et al. [113] addressed the sensitivity feature of the UHF RFID sensors and investigated their potential as indoor environmental corrosiveness detectors sensitive to variation of the metal thickness loss in the range of tens of nanometers. One of the controversial issues for corrosion monitoring sensors is the interpretation of the output data when localized corrosion phenomena appear. Yasri et al. [114] proposed a novel RF-based technique for atmospheric corrosion monitoring, the results of which showed the ability to distinguish between uniform and localized corrosion mechanisms. Another important usage for this technique within corrosion monitoring applications is in tracking the condition of the coating layer and monitor its degradation process [115]. This ability allows for the improvement of infrastructure maintenance operations and can result in large financial savings in many industrial fields. Generally, the basic components of an RFID system are a reader and a tag sensor. The tag is a transponder with integrated circuits and an antenna that can be activated by the reader’s electromagnetic radiation, and it communicates stored information back. Figure 14 shows a detailed view of a commercial RFID system´s scheme for crack detection. Depending on the power supply, tags can be active or passive. Active tags can convey data and have their own power sources (such as a battery or solar cell), making them larger and more expensive. Passive tags convey data in response to reader inquiry, since their only power source is the electromagnetic field transmitted by the reader. These tags are simpler, less expensive, more dependable, and more widely utilized. Therefore, for low-cost corrosion monitoring applications, passive wireless RFID sensors are predominantly used. In the publications reviewed in this study, Lippincott [116], Dante et al. [117], Palmer et al. [118], He et al. [119], Zhang et al. [120], Sunny et al. [121], Bruciati et al. [122], and Marindra et al. [123] addressed the use of low-cost passive wireless RFID sensors (tags) in order to detect physical parameters such as coating thickness and crack propagation which are essential corrosion elements in pipelines and concrete structures. It is important to mention that in the aforementioned studies, real-time corrosion monitoring and corrosion rate calculation were not performed. However, Zhang et al. [124], Zarifi et al. [125], Deif et al. [126], and Rioual et al. [127] developed RFID-based low-cost corrosion monitoring systems with real-time update ability.

#### 3.3.3. Optical Fiber

Over the last few decades, optical fibers have been developed and widely used for a variety of applications in the oil and gas sector; civil engineering construction; structural health monitoring; and cracks, strain, temperature, and pH monitoring. Essentially, the approach employs optical fibers composed of glass or plastic that convey light along their length and are thus sensitive to any parameter that can change the intensity, frequency, polarity, or phase of the light. For corrosion monitoring, optical fiber sensors such as fiber Bragg gratings (FBG), interferometers, surface plasmon resonances (SPR), distributed sensing, or optical intensity modulations are used. As can be seen in Figure 15, optical fiber sensors for low-cost corrosion monitoring applications are categorized into four types based on how they detect the presence of corrosion in the structure. The first group employs the direct measurement of corrosion effects. For that purpose, sensors measure parameters such as corrosion-induced strain, deformation, and displacements within the structure. FBGs, white light interferometry, and optical time domain interferometry (OTDR)—a technology within optical distributed sensing—are among the optical sensors used in this category. These sensors are implanted in the building’s structure in order to monitor the corrosion in concrete structures as well as in civil and military airframes where corrosion is a worry because they are working beyond their projected lifetime in some circumstances.

Among the authors who developed low-cost corrosion monitoring solutions based on optical fiber sensors, Zhao et al. [97] implemented a white light interferometer, Ramani et al. [128] addressed the use of lens-based plastic optical fibers (LPOF), and Dos Santos et al. [129] and Chen et al. [130] employed long-period grating (LPG) sensors to detect and monitor parameters such strain and corrosion crack propagation in their projects. In the second group, the corrosion of the structure of interest was estimated rather than directly measured by monitoring the corrosion of a similar substance deposited onto an optical fiber. Obviously, this sensor must be situated near the structure whose corrosion is being investigated in order to experience identical conditions. Wan et al. [131] and Guan et al. [132] employed this approach to monitor corrosion in metals such as steel and aluminum as well as for the monitoring of coating layer thickness. In terms of corrosion precursors, optical fiber sensors can detect ions that cause a higher corrosion rate, with waterborne chlorides being the most relevant. Elster et al. [133] and Chen et al. [134] used fluorescence-based optical fiber sensors to detect chlorides ions inside of concrete structures.

Environmental parameters such as humidity and pH play an important role in corrosion, therefore optical fiber sensors such as fluorescence or colorimetry are widely used for monitoring these elements. Cooper et al. [135], Casas et al. [136], and Kiremidjian et al. [137] developed novel low-cost systems based on the combination of type 1 and type 4 measurements using LPG sensors in order to detect both the strain and environmental parameters and therefore obtain better corrosion data on the analyzed structure.

#### 3.3.4. Ultrasonic Wave

Ultrasonic corrosion monitoring is a type of non-destructive testing (NDT) that is used to track the corrosion process. High-frequency acoustic waves (sound waves) are used in this monitoring technique to measure or map the interior structure, thickness, and other properties of the material being monitored. There are numerous commercial ultrasound instruments available that provide useful information regarding various types of degradation (i.e., cracks, deformations, thinning, and corrosion). They are, however, generally classified as inspection equipment rather than monitoring devices. Therefore, in this this paper, publications that presented low-cost corrosion monitoring solutions for various industries based on ultrasound technology are reviewed. For instance, in the aerospace industry, the use of ultrasonic sensors is very common. Zhao et al. [138], Meyendorf et al. [139], and Perelli et al. [140] introduced novel low-cost monitoring systems based on ultrasound technology for aircraft´s health monitoring applications. In the oil industry, in order to track the health of the pipelines, Wan et al. [141] and Barshinger et al. [142] developed low-cost wireless ultrasound-based monitoring devices. Harper et al. [143] and Jalali et al. [144] proposed remote ultrasound-based techniques for corrosion monitoring in civil structures. The aforementioned corrosion monitoring systems take highly frequent readings in order to detect minor variations, which in this case are connected to corrosion phenomena. The accuracy of ultrasound signals enables very frequent thickness measurements that can be used to monitor corrosion.

#### 3.3.5. Electromagnetic Induction

Over the last few years, a novel corrosion control method has been created for protecting damaged, painted surfaces in contact with atmospheric conditions, and it employs electromagnetically induced surface currents. In addition, this technique has attracted considerable attention for the initial monitoring of corrosion in concrete structures. Several scholars have introduced monitoring sensors based on Faradays law of electromagnetic induction (EMI) for the early assessment of corrosion in reinforced concrete (RC) structures. In this review, authors such as Andringa et al. [145], Vinogradov [146], Chen et al. [147,148], and Kranz et al. [149] introduced low-cost corrosion monitoring solutions based on this technique. The monitoring devices used in the aforenamed experiments are made up of multiple loop coils (MLCs) which operate as sensors and have a receiver and a transmitter coil. The positioning of the two coils on concrete structures measures the potential difference, which is then used to analyze the extent of the corrosion. Gallo et al. [150] introduced a novel corrosion monitoring method based on the magnetic field induced by the electrochemical activity from a uniform corrosion mechanism of a metal sample. Figure 16 shows a scheme of the monitoring system containing a two-unit set of commercial giant magneto-resistive (GMR) sensors for tracking the magnetic fields from the active corrosion of metal plates.

#### 3.3.6. Image Recognition

Human-based visual inspection, which is time-consuming, costly, and even dangerous, has so far been one of the most-used techniques for structural health monitoring applications. Concerns about safety, high costs, as well as recent developments in image detection sensors and AI technology have highlighted the urge for the development of a reliable, low-cost, safe, and quantitative solution known as image recognition (IR) or image analysis technology. In fact, the phenomenon of corrosion leads to changes in the surface structure, morphology, and composition of materials [151]. These changes can be captured by a number of image sensors, and by implementing image recognition technologies, information about the type and extent of corrosion can be obtained [152]. Figure 17 illustrates a simple image-collection setup using a portable CCD camera. Xia et al. [153] presented a comprehensive review of the digital image analysis systems as well as the developed algorithms used to assess and process the corrosion images. In addition, some physical and electrochemical methods that can be employed to support the IR results are summarized in this study. However, this technique is still considered to be new for low-cost corrosion monitoring applications, and it has not yet been widely investigated. Among the reviewed publications, Igoe et al. [154] addressed the use of smartphone cameras to detect and monitor the corrosion process on the surface of a clean iron sample. Smartphone technology offers numerous prospects for increased participation in scientific and technological research. In fact, in this research, congruent validation tests and errors of less than 5% demonstrated the efficiency of this method for the characterization of a smartphone image sensor response to the degree of steel corrosion. These findings show that a smartphone can be used to evaluate surface corrosion efficiently and at a low cost. Moreover, Jindal et al. [155] proposed the use of a panoramic image transmission and refinement technique in order to overcome difficulties such high time consumption, blurred data, and color variations that are common problems in underwater pipeline monitoring applications. The proposed method was validated efficiently in terms of energy consumption and communication, and it showed maximum data delivery at a low-cost compared to conventional techniques. A more recent development in the field of corrosion monitoring using image sensors was produced by Benkhoui et al. [156]. In this study the authors introduced the use of an unmanned aerial vehicle (UAV) for crack detection in a bridge and investigated the adoption of stereo cameras based on passive and active depth calculation for the assessment of bridge structural integrity.

## 4. Conclusions

In this paper, a systematic literature review of low-cost technologies for corrosion monitoring applications has been presented. After the filtering process, 90 articles published since 1986 were examined that address the use of low-cost technologies such as sensors and microcontrollers in the corrosion monitoring field. According to the reviewed articles, this paper could provide the readers with an overview of the low-cost technologies that have been implemented so far for the monitoring of corrosion parameters using electrochemical methods (EIS, LPR, galvanic, and EN) and physical methods (ER, RF, OF, UW, EMI, and IR). In the case of using electrochemical methods as the principal measurement technique, it was found that the majority of the authors (58%) focused their studies on the use of EIS, as it turned out to be the most reliable and feasible method for corrosion monitoring applications. In the case of physical methods, ER has proven to be the most reliable method, indicating that 25% of the scholars were focused on development of their respective low-cost corrosion monitoring systems based on this technique. Regarding the architecture of the low-cost data acquisition system, it has been discovered that the Arduino microcontroller has attracted a lot of attention for EIS and ER measurements. Moreover, among the physical methods, it was found that RFID and OF sensors were the two second-most-reliable areas in which the majority of the researchers focused the development of their respective low-cost corrosion monitoring systems (23% and 21%, respectively). Furthermore, it has been observed that most of the reviewed articles were published in the US (43 out of 90), and the majority of the projects were established in the civil structure industry (46 out of 90).

This article shows that, despite the fact that a significant number of studies have been reviewed with the goal of employing low-cost technologies for corrosion monitoring, there are still some gaps in the literature that must be addressed in future research. The same systematic literature research should be conducted using other databases (such as IEEE explore or Google Scholar) in order to give a full analysis of the used low-cost solutions for corrosion monitoring. It is also necessary to provide an overview of the existing low-cost sensors for corrosion monitoring in the market and compare their features with the those in the literature.

## Figures and Tables

**Figure 1 sensors-23-01309-f001:**
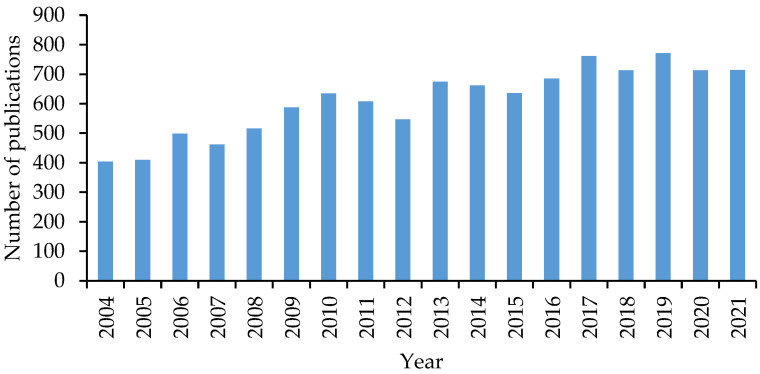
Distribution of publications on corrosion monitoring over time (2004–2021) in the SCOPUS database.

**Figure 2 sensors-23-01309-f002:**
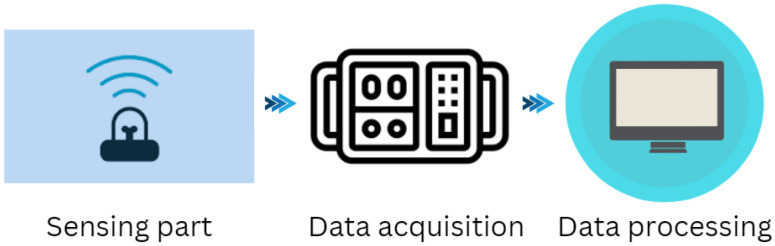
Scheme of a corrosion monitoring system.

**Figure 3 sensors-23-01309-f003:**
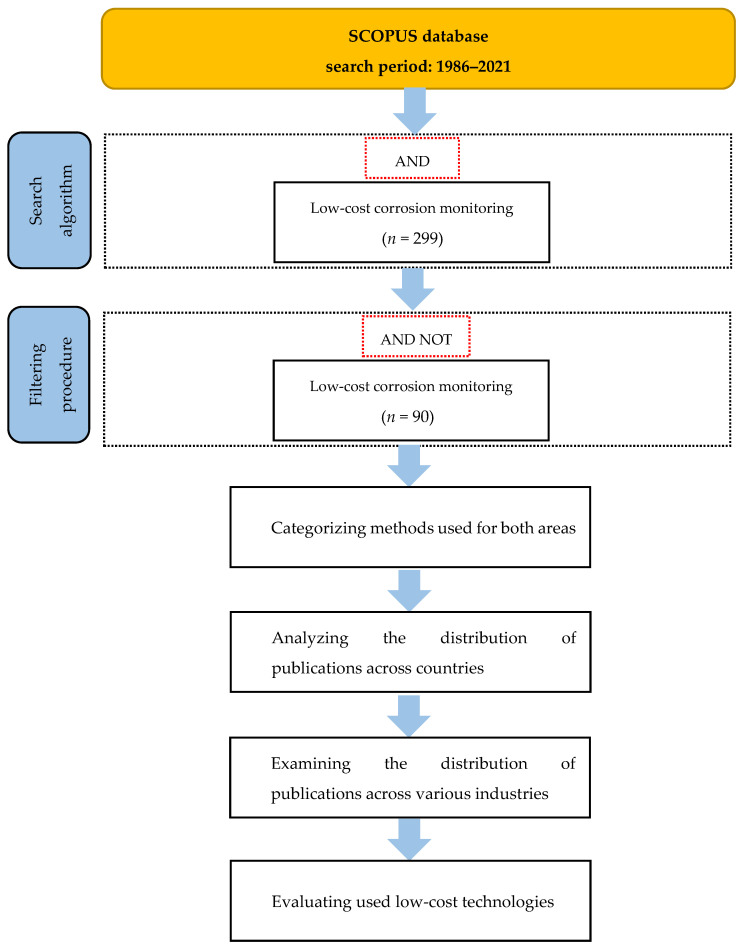
Flowchart depicting the systematic data sampling procedure.

**Figure 4 sensors-23-01309-f004:**
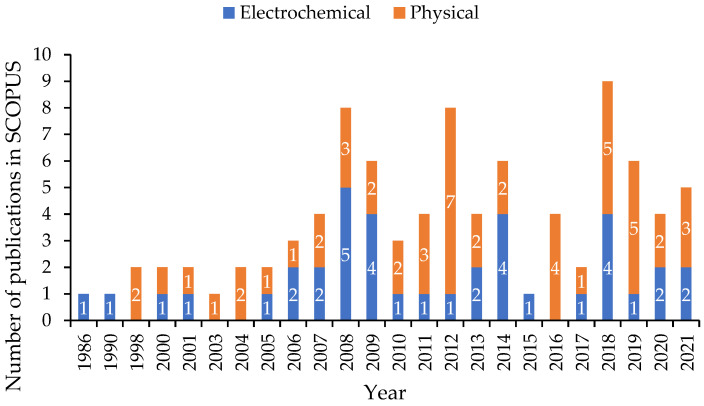
Distribution of publications over time (1986–2021) from the SCOPUS database.

**Figure 5 sensors-23-01309-f005:**
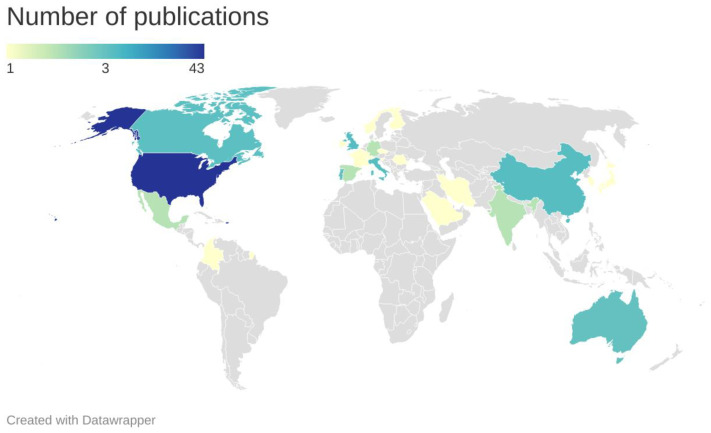
Distribution of the publications on low-cost corrosion monitoring by country.

**Figure 6 sensors-23-01309-f006:**
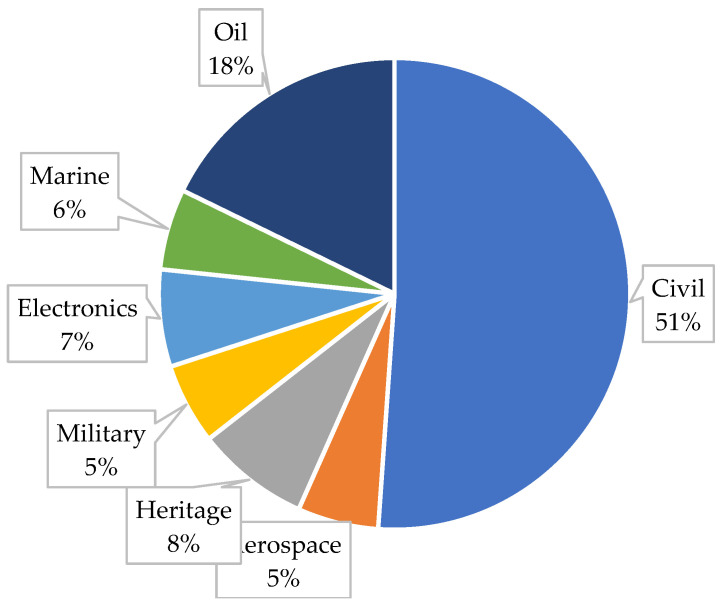
Contribution of the publications on low-cost corrosion monitoring by industry.

**Figure 7 sensors-23-01309-f007:**
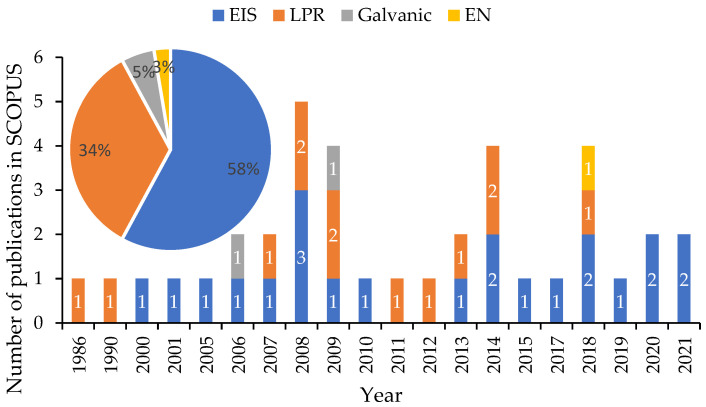
Distribution of the reviewed articles associated with the application of electrochemical methods for corrosion monitoring from 1986 to 2021.

**Figure 8 sensors-23-01309-f008:**
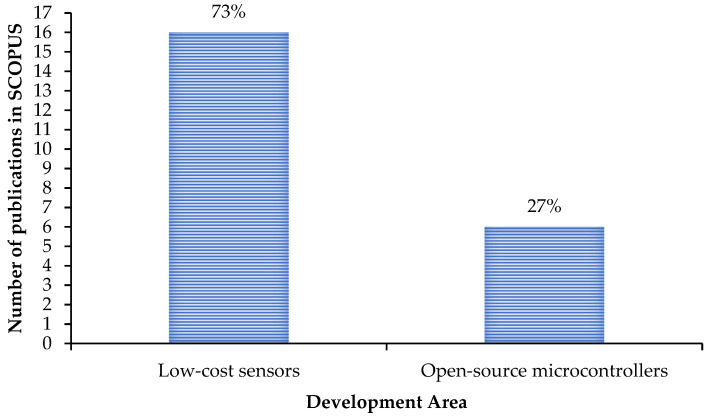
Two areas of development in the SCOPUS database for the application of low-cost corrosion monitoring solutions using the EIS technique: Area 1 focusing on the development of the sensing elements, and Area 2 focusing on the use of open-source microcontrollers as the low-cost data acquisition system.

**Figure 9 sensors-23-01309-f009:**
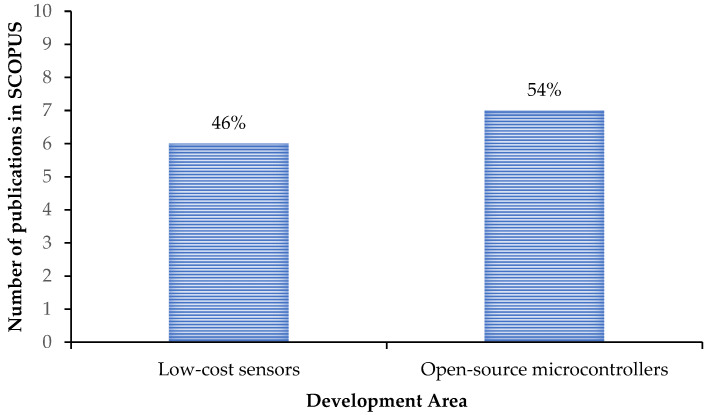
Two areas of development in the SCOPUS database on the application of low-cost corrosion monitoring solutions using the LPR technique: Area 1 focusing on the development of the sensing elements, and Area 2 focusing on the use of open-source microcontrollers as the low-cost data acquisition system.

**Figure 10 sensors-23-01309-f010:**
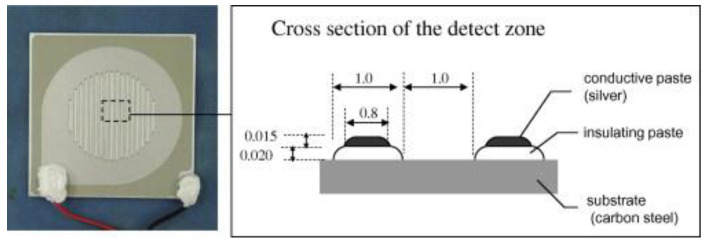
Fe–Ag-type atmospheric corrosion monitor (ACM) sensor (reproduced with permission from Reference [26]).

**Figure 11 sensors-23-01309-f011:**
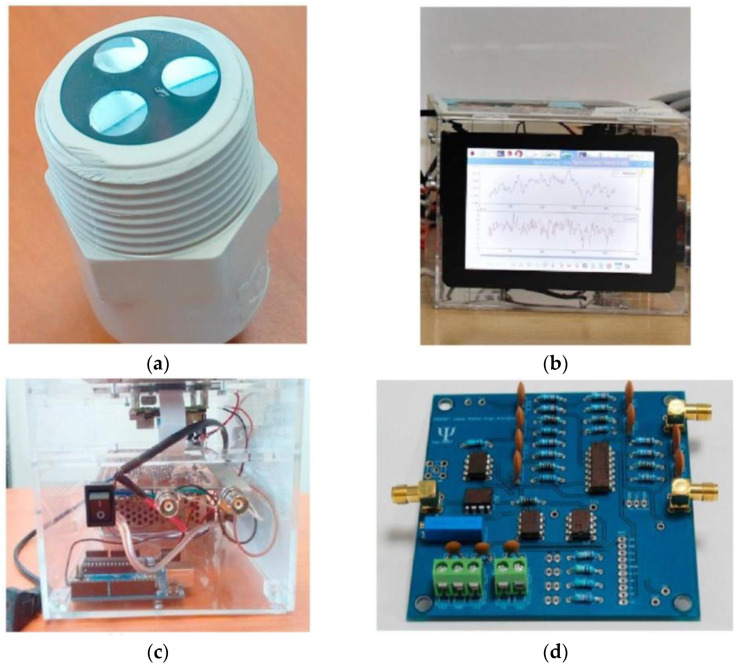
The developed low-cost EN-based corrosion monitoring system: (**a**) three-electrode probe made of aluminum; (**b**) touchscreen display for communication and graphical interface; (**c**) on–off switch and connectors of the portable device; (**d**) the electronic circuit board for the signal amplification (reproduced with permission from Reference [27]).

**Figure 12 sensors-23-01309-f012:**
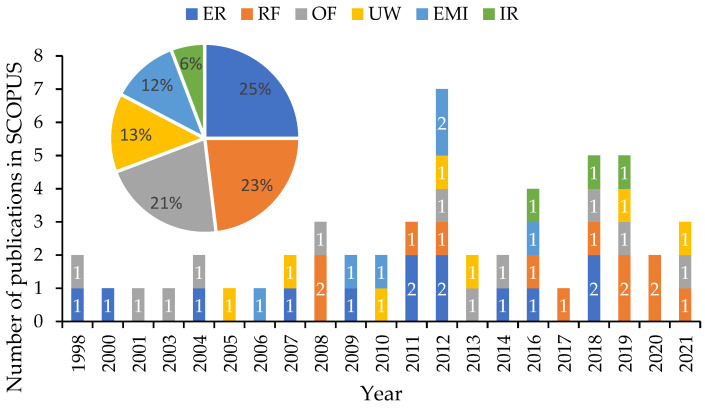
Distribution of the reviewed articles associated with the application of physical methods for corrosion monitoring from 1986 to 2021.

**Figure 13 sensors-23-01309-f013:**
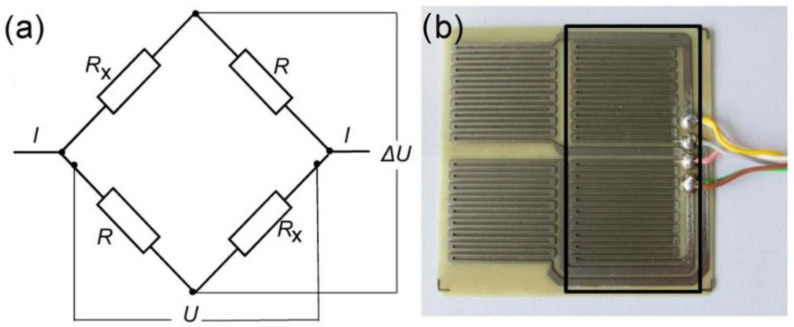
An ER sensor made of high-strength steel with four elements: (**a**) a schematic illustration of the sensor; (**b**) a photo of the sensor, protected by a transparent epoxy coating (reproduced with permission from Reference [24]).

**Figure 14 sensors-23-01309-f014:**
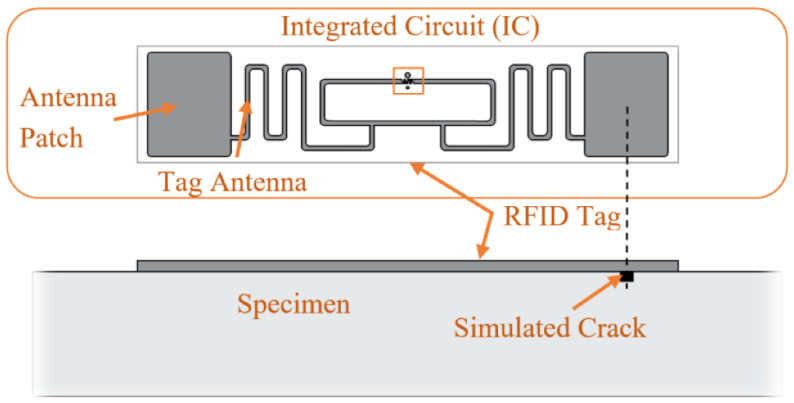
The commercial UHF RFID tag ALN-9662 (reproduced with permission from Reference [122]).

**Figure 15 sensors-23-01309-f015:**
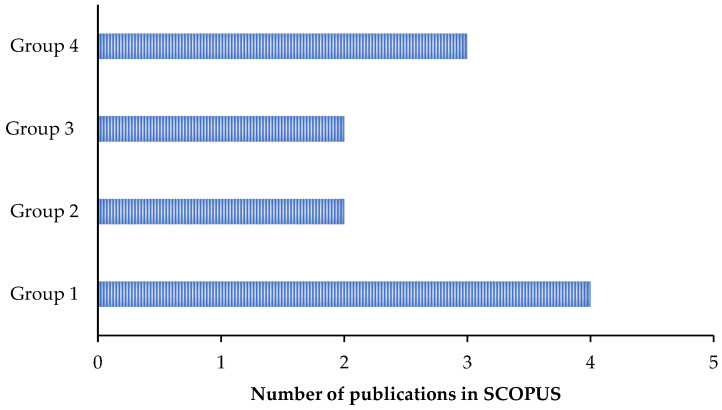
Distribution of publications in the SCOPUS database between four areas of corrosion measurement using optical fiber sensors.

**Figure 16 sensors-23-01309-f016:**
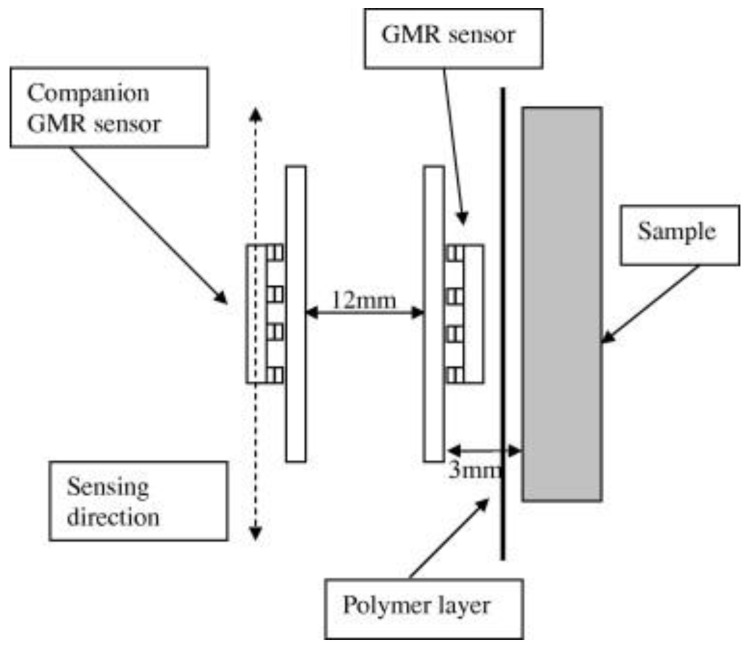
Scheme of the side view of the sensor setup (reproduced with permission from Reference [150]).

**Figure 17 sensors-23-01309-f017:**
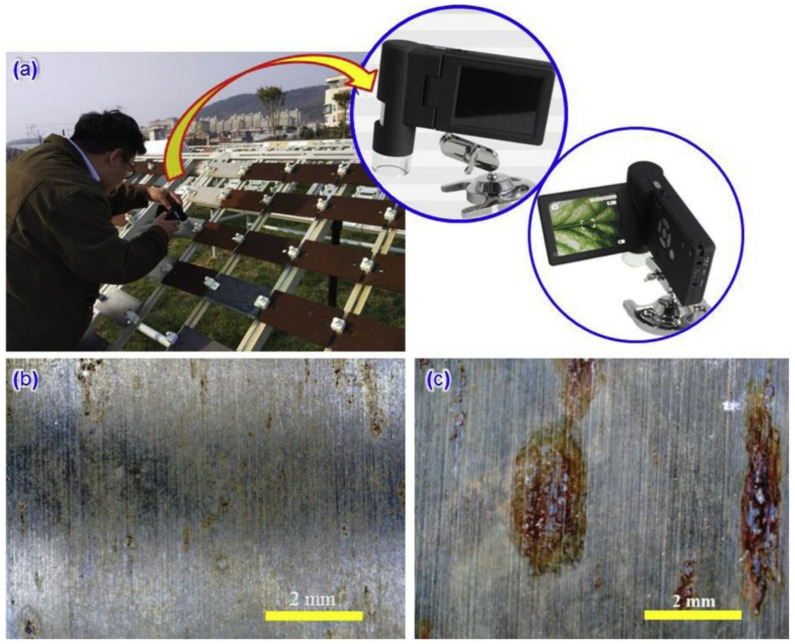
Collection of corrosion images of metallic materials in the field using a portable CCD camera: (**a**) image collection, (**b**,**c**) corrosion images of 304 SS exposed to a marine atmosphere (reproduced with permission from Reference [153]).

**Table 1 sensors-23-01309-t001:** Comparison of atmospheric corrosion methods.

Method	Sensitivity	Benefits	Drawbacks
Coupons	Medium, unless there are long exposure times	Standardized method Simple to analyze the data	Does not have real-time updates Requires a long time for measurements
Galvanic	Medium	Not affected by temperature changes Suitably equipped for harsh outdoor conditions	Galvanic corrosion increases the corrosion rate Electrolyte connection required
EIS	Medium	Details of the mechanism of corrosion Non-destructive examination of coatings	Data interpretation requires deep knowledge of EIS Unevaluable data in the presence of a thick corrosion product Electrolyte connection required
EN	Medium	Localized detection Identification of the corrosion mechanism	Complicated data interpretation Electrolyte connection required
ER	High	Standardized method Very sensitive Data interpretation and operation are simple. The best option for consistent corrosion monitoring	Temperature dependent Monitoring limitations for non-uniform corrosion

**Table 2 sensors-23-01309-t002:** List of commercial corrosion sensors in the market for real-time update.

Technique	Device	Corrosion Rate Range (mpy)	Price ($US)	Ref.
ER	ZK9800	0.00394–39.34	1250	[24]
EIS	CST1808	0.00394–0.3934	5511	[25]
Galvanic	CST480MS	0.00394–3.934	3150	[26]
LPR	AquaMate	0.01–200	2925	[27]

**Table 3 sensors-23-01309-t003:** Shows methods for the two main fields of electrochemical and physical techniques.

Fields of Corrosion Monitoring	Electrochemical				Physical					
Methods used	EIS	LPR	Galvanic	EN	ER	Optical fiber	Radio frequency	Image sensor	Ultrasonic	Electromagnetic induction

## Data Availability

Not applicable.

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
