# Peer review of "Low-Cost Technologies Used in Corrosion Monitoring"

_sensors, 2023, doi:10.3390/s23031309_

Round 1

Reviewer 1 Report

Low-Cost Technologies Used in corrosion Monitoring, this is a review paper, the author has cited a many relevent  references, overall the paper is very interesting, and the topic is difinitely important for corrosion science and engineering.  I suggest a minor revision of the work:

1. The figures shown in this review paper should be presentitive and show typical results in the field. I suggest 1-2 figures should be added, to highlight the most important sensors.

2. image analyses is also an Low-Cost Technologies Used in Corrosion Monitoring, as can be seen in this paper: Material degradation assessed by digital image processing Fundamentals, progresses, and challenges.   The research group in Tianjin University has done lots of work in this field.

3. Several important work is not refered, the reviewer suggest the author read the following papers:

[1] Electrochemical probes and sensors designed for time-dependent atmospheric corrosion monitoring Fundamentals, progress, and challenges

[2] Electrochemical Noise Applied in Corrosion Science Theoretical and Mathematical Models towards Quantitative Analysis

[3] Detection of atmospheric corrosion of aluminum alloys by electrochemical probes theoretical analysis and experimental tests

Author Response

Response to Reviewers’ comments and description of changes in the revised Manuscript ID sensors-2132762, submitted to Physical Sensors

Authors’ general response to Reviewer comments:

The authors are grateful to the Reviewers for the time dedicated to revising our paper and the provided positive feedback and comments. We have done our best to implement all suggested changes to the manuscript and we are certain this has helped improve its quality. Our responses to the review comments are below, and changes that we have made to the paper are highlighted in the revised manuscript submission.

Reviewer #1:

Low-Cost Technologies Used in corrosion Monitoring, this is a review paper, the author has cited a many relevant references, overall the paper is very interesting, and the topic is definitely important for corrosion science and engineering.  I suggest a minor revision of the work:

Point 1: The figures shown in this review paper should be presentative and show typical results in the field. I suggest 1-2 figures should be added, to highlight the most important sensors.

Response 1:

Authors agree that descriptive figures for developed low-cost sensors will add a significant value to the current study. However, due to the fact that the main purpose of this paper was to present a systematic literature review about the use of low-cost digital technologies in this field only comparative figures and charts were shown previously. Considering, the reviewer suggestion, authors have added some figures to highlight the schematic description of the most important sensors used in the literature.

Notes/actions:

Figure 10 (line 502-504), Galvanic or ACM sensor

 Figure 11 (lines 524-528), EN-based monitoring system

Figure 13 (lines 600-603), ER sensor

Figure 14 (lines 641-642), RFID sensor

Figure 16 (lines 722-723) and EMI-based sensing scheme

Figure 17 (lines 757-760), Image analysis method

Point 2: Image analyses is also a Low-Cost Technologies Used in Corrosion Monitoring, as can be seen in this paper: Material degradation assessed by digital image processing Fundamentals, progresses, and challenges. The research group in Tianjin University has done lots of work in this field.

Response 2:

Authors agree to the fact that image analysis is indeed considered to be one of the most recent low-cost solutions for corrosion monitoring and in the manuscript in section 3.3.6, this technique was introduced as Image Recognition technology. However, the aforementioned review paper: [1] Material degradation assessed by digital image processing Fundamentals, progresses, and challenges, has been reviewed by authors and very impressive information was obtained. Furthermore, it was cited in the manuscript. In addition, the following papers from Tianjin University research group has been cited:

[2] Image analysis of atmospheric corrosion of field exposure high strength aluminium alloys

[3] Atmospheric corrosion assessed from corrosion images using fuzzy Kolmogorov–Sinai entropy

Notes/actions:

[1] Reference number: [153] was cited in line 734,

[2] Reference number: [151] was cited in line 731,

[3] Reference number: [152] was cited in line 733,

Lines 730-738 and figure 17 (lines 757-760) were added.

Point 3: Several important works are not referred; the reviewer suggests the author read the following papers:

[1] Electrochemical probes and sensors designed for time-dependent atmospheric corrosion monitoring Fundamentals, progress, and challenges

[2] Electrochemical Noise Applied in Corrosion Science Theoretical and Mathematical Models towards Quantitative Analysis

[3] Detection of atmospheric corrosion of aluminum alloys by electrochemical probes theoretical analysis and experimental tests.

Response 3:

The authors agree with the reviewer about the convenience of adding the aforementioned works. For that reason, authors have read and cited mentioned papers in the manuscript.

Notes/actions:

[1] Reference number: [12] cited in line 125.

[2] Reference number: [96] cited in line 512 & lines 507-512 were added.

[3] Reference number: [21] cited in line 152.

Reviewer 2 Report

This paper is a review of low-cost sensors for corrosion monitoring. It is not so easy since corrosion is a complex phenomenon and many different sensors were proposed recently. In the introduction, the authors report the different techniques currently used for corrosion monitoring. As explained (Table 2), it is of first importance to realize sensors with reasonable price. It is indeed a challenging task. The authors used the term “Low-cost corrosion monitoring” to extract the references from the literature. The term “low-cost” is debatable since it depends on the financial funds available by the end-user and it is therefore not always used to define the proposed innovative technical solutions. The term “low-cost” is therefore also a source of discussion between authors and reviewers and is frequently suppressed in the manuscripts This is in particularly the case of RFID, microwave or radio frequency sensors which are implicitly “low-cost” by the use of this technology. Consequently, the references which appear in the manuscript are rather limited with respect to the numerous works dedicated to this topic. A list of recent papers note cited in this review is given below. I suggest the authors to include them in the manuscript and to add the references associated with these papers (cited and citing). Regarding the high number of references, it is also important to give furthers details on these papers (lines 618/637) and not to restrict the discussion with 15 lines. ER sensors are integrated in the part “Electrochemical sensors”. They should be placed in “physical sensors” since they are not based in electrochemistry. Additional reference should be included (see for example the paper : Corrosion monitoring in atmospheric conditions : a review by K. Popova).

UHF-RFID corrosion sensor in concrete :

Development of an embedded UHF-RFID corrosion sensor for monitoring corrosion of steel in concrete;  K. Bouzaffour, B. Lescop, F. Gallée, P. Talbot, and S. Rioual, IEEE Sensors Journal 21 12306-12312 (2021).

Sensitivity of UHF-RFID corrosion sensors:

Development of a RFID sensitive tag dedicated to the monitoring of the environmental corrosiveness for indoor applications; I.El Masri, B. Lescop, P. Talbot, G.N. Vien, J. Becker, D. Thierry, S. Rioual, Sensors and Actuators: B. Chemical 322 128602 (2020).

Separating localised and uniform corrosion : an issue for corrosion sensors

Monitoring uniform and localised corrosion by a radiofrequency sensing method ; M. Yasri, B. Lescop, E. Diler, F. Gallée, D. Thierry and S. Rioual, Sens. Actuators B : Chem.,257 988-992 (2018).

Degradation of organic coating before corrosion of the underliying metal:

Development of a Radio Frequency resonator for monitoring water diffusion in organic coatings ; R. Khalifeh, B. Lescop, F. Gallée, and S. Rioual, Sens. Actuators A: Phys. 247, 30-36 (2016)

Author Response

Response to Reviewers’ comments and description of changes in the revised Manuscript ID sensors-2132762, submitted to Physical Sensors

Authors’ general response to Reviewer comments:

The authors are grateful to the Reviewers for the time dedicated to revising our paper and the provided positive feedback and comments. We have done our best to implement all suggested changes to the manuscript and we are certain this has helped improve its quality. Our responses to the review comments are below, and changes that we have made to the paper are highlighted in the revised manuscript submission.

Reviewer #2:

Point 1: This paper is a review of low-cost sensors for corrosion monitoring. It is not so easy since corrosion is a complex phenomenon and many different sensors were proposed recently. In the introduction, the authors report the different techniques currently used for corrosion monitoring. As explained (Table 2), it is of first importance to realize sensors with reasonable price. It is indeed a challenging task.

Response 1:

In this section the main purpose of the authors was to emphasize on the fact that the current commercial corrosion monitoring systems are considered to be expensive for structural health monitoring application in civil structures (atmospheric corrosion) compared to other common sensors for SHM applications (such as accelerometers, inclinometers, thermal, humidity and strain gauge sensors). However, in order to include more sensors, authors included another common corrosion monitoring sensor (LPR-based) found in the literature.  

Notes/actions:

Table 2 is updated.

Point 2: The authors used the term “Low-cost corrosion monitoring” to extract the references from the literature. The term “low-cost” is debatable since it depends on the financial funds available by the end-user and it is therefore not always used to define the proposed innovative technical solutions. The term “low-cost” is therefore also a source of discussion between authors and reviewers and is frequently suppressed in the manuscripts. This is in particularly the case of RFID, microwave or radio frequency sensors which are implicitly “low-cost” by the use of this technology. Consequently, the references which appear in the manuscript are rather limited with respect to the numerous works dedicated to this topic. A list of recent papers not cited in this review is given below. I suggest the authors to include them in the manuscript and to add the references associated with these papers (cited and citing). Regarding the high number of references, it is also important to give further details on these papers (lines 618/637) and not to restrict the discussion with 15 lines.

UHF-RFID corrosion sensor in concrete :

[1] Development of an embedded UHF-RFID corrosion sensor for monitoring corrosion of steel in concrete;  K. Bouzaffour, B. Lescop, F. Gallée, P. Talbot, and S. Rioual, IEEE Sensors Journal 21 12306-12312 (2021).

Sensitivity of UHF-RFID corrosion sensors:

[2] Development of a RFID sensitive tag dedicated to the monitoring of the environmental corrosiveness for indoor applications; I.El Masri, B. Lescop, P. Talbot, G.N. Vien, J. Becker, D. Thierry, S. Rioual, Sensors and Actuators: B. Chemical 322 128602 (2020).

Separating localised and uniform corrosion : an issue for corrosion sensors

[3] Monitoring uniform and localised corrosion by a radiofrequency sensing method ; M. Yasri, B. Lescop, E. Diler, F. Gallée, D. Thierry and S. Rioual, Sens. Actuators B : Chem.,257 988-992 (2018).

Degradation of organic coating before corrosion of the underliying metal:

[4] Development of a Radio Frequency resonator for monitoring water diffusion in organic coatings ; R. Khalifeh, B. Lescop, F. Gallée, and S. Rioual, Sens. Actuators A: Phys. 247, 30-36 (2016)

Response 2:

Authors acknowledge that RFID sensors are considered to be one of the low-cost methods available for corrosion monitoring applications. That is why several studies in this field was reviewed in the subsection 3.3.2 (Radio Frequency). It is important to mention that the main goal of this systematic literature review was to include those publications that introduced or developed a novel low-cost system using inexpensive technologies (such as single board computers, electronic circuits, microcontrollers and sensors) for each electrochemical and physical method available to make them even more cost effective. However, authors appreciate your suggestion and accept to extend subsection 3.3.2 giving more detail about this method and cite the aforementioned papers.

Notes/actions:

[1] reference number: [112] cited in line 609 and lines 609-611 were added.

[2] reference number: [113] cited in line 611 and lines 611-614 were added.

[3] reference number: [114] cited in line 616 and lines 614-618 were added.

[4] reference number: [115] cited in line 620 and lines 618-621were added.

Lines 638-640 were edited.

In addition, figure 14, a scheme of a commercial UHF RFID tag was added (lines 641-642)

Figure 14. The commercial UHF RFID tag ALN-9662 (Reproduced with permission from [122]).

Point 3: ER sensors are integrated in the part “Electrochemical sensors”. They should be placed in “physical sensors” since they are not based in electrochemistry.

Response 3:

Authors agree that ER sensors should be included in the physical sensors section, due to the fact that they only measure electrical resistance and furthermore thickness loss and corrosion rate values are calculated and no electrochemical measurements are applied using these sensors in atmospheric corrosion phenomenon. Therefore, major changes have been applied to the figures and the manuscript.

Notes/actions:

Table 3, figures 4,7 and 12 have been entirely modified. Section 3.2.2 (ER) has been moved to section 3.3.1 in the physical method section. Lines 343-352, 388, 390, 392, 396, 398, 402-405, 406, 533, 535-535, 540, 542-544, 546, 549-551, 553, 556-561 and 568 were modified.

Point 4: Additional reference should be included (see for example the paper: Corrosion monitoring in atmospheric conditions: a review by K. Popova).

Response 4:

Mentioned reference (Corrosion monitoring in atmospheric conditions: a review) was already cited in this paper. The reference number is [13].

Notes/actions:

This reference has been cited in lines 129, 219 and 408.

Reviewer 3 Report

The manuscript deals with a very interesting topic regarding corrosion monitoring. The work is of a review nature.

1. However, a major shortcoming of the article is the lack of any description of exemplary specific solutions of sensors or measuring devices. It would be necessary to add good drawings of sensors in each subsection, such as 3.2.1., etc., and comment on the example shown.

2. The second significant lack in the article is the omission of the description of the phenomenon of corrosion, its various varieties. It is particularly important to separately describe the mechanism on structural steel coated with an anti-corrosion coating and the corrosion of reinforcing steel in concrete, as well as the corrosion of the concrete itself.

Author Response

Response to Reviewers’ comments and description of changes in the revised Manuscript ID sensors-2132762, submitted to Physical Sensors

Authors’ general response to Reviewer comments:

The authors are grateful to the Reviewers for the time dedicated to revising our paper and the provided positive feedback and comments. We have done our best to implement all suggested changes to the manuscript and we are certain this has helped improve its quality. Our responses to the review comments are below, and changes that we have made to the paper are highlighted in the revised manuscript submission.

Reviewer #3:

The manuscript deals with a very interesting topic regarding corrosion monitoring. The work is of a review nature.

Point 1: However, a major shortcoming of the article is the lack of any description of exemplary specific solutions of sensors or measuring devices. It would be necessary to add good drawings of sensors in each subsection, such as 3.2.1., etc., and comment on the example shown.

Response 1:

Authors agree that descriptive figures for developed low-cost sensors will add a significant value to the current study. However, due to the fact that the main purpose of this paper was to present a systematic literature review about the use of low-cost digital technologies in this field only comparative figures and charts were shown previously. Considering, the reviewer suggestion, authors have added some figures to highlight the schematic description of the most important sensors used in the literature.

Notes/actions:

Figure 10 (line 502-504), Galvanic or ACM sensor

 Figure 11 (lines 524-528), EN-based monitoring system

Figure 13 (lines 600-603), ER sensor

Figure 14 (lines 641-642), RFID sensor

Figure 16 (lines 722-723) and EMI-based sensing scheme

Figure 17 (lines 757-760), Image analysis method

Point 2: The second significant lack in the article is the omission of the description of the phenomenon of corrosion, its various varieties. It is particularly important to separately describe the mechanism on structural steel coated with an anti-corrosion coating and the corrosion of reinforcing steel in concrete, as well as the corrosion of the concrete itself.

Response 2:

It is important to mention that the main goal of this paper is to review low-cost techniques used for corrosion monitoring applications to make this process cost effective, especially for those scholars who are interested to employ corrosion sensors for structures that don´t have high structural health monitoring budgets. To do this, the most common electrochemical and physical corrosion monitoring methods that emphasized on the use of inexpensive digital technologies (such as single board computers, electronic circuits and microcontrollers) or a novel design of the sensing elements (to make them cheaper) to make those methods cost effective, are reviewed. In addition, this paper does not aim to provide detailed information about the type of corrosion occurring in different systems, because it could increase the length of the paper and would exceed the limits of the journal. Therefore, at the beginning of each subsection a brief introduction including the most important references about each measurement method was given which scholars could refer for a general understanding over the corrosion phenomenon and its different types. In fact, reviewing the sensors based on the type of corrosion is one of the future goals of the authors for a future publication. However, in order to follow the reviewer´s suggestion, authors have added a breif explanation about the type of corrosion (such as localized and uniform corrosion for metallic parts, degradation of the coating layer and concrete corrosion) that the corresponding sensor is able to cover, for each method.

Notes/actions:

Subsection 3.2.1: line 417: coating tracking and monitoring

Subsection 3.2.4: lines 507-509 mentioning type of corrosion covered by this method.

Subsection 3.3.1: line 584 showing that these sensors are for uniform corrosion.

Subsection 3.3.2: lines 609-621 summarizing the applications of this technique such as: corrosion in concrete, separating localized and uniform corrosion, degradation of organic coating and indoor environmental corrosivity monitoring.

Subsection 3.3.5: lines 709-711: corrosion in concrete & lines 717-721 uniform corrosion.

Round 2

Reviewer 2 Report

The changes were made according to my remarks.

Reviewer 3 Report

I welcome the additions to the manuscript of drawings and texts. In its present form, I consider the article fit for publication.